# Stress burden related to postreperfusion syndrome may aggravate hyperglycemia with insulin resistance during living donor liver transplantation: A propensity score-matching analysis

**Sumin Chae[1], Junghee Choi[2], Sujin Lim[2], Ho Joong Choi[3], Jaesik Park[2], Sang Hyun Hong[2], Chul Soo Park[2], Jong Ho Choi[2], Min Suk Chae[2]** *

1 Department of Anesthesiology and Pain Medicine, United Hospital, Seoul, Republic of Korea,
2 Department of Anesthesiology and Pain Medicine, Seoul St. Mary's Hospital, College of Medicine, The Catholic University of Korea, Seoul, Republic of Korea, 3 Department of Surgery, Seoul St. Mary's Hospital, College of Medicine, The Catholic University of Korea, Seoul, Republic of Korea

* shscms@gmail.com

## Abstract

### Background

We investigated the impact of postreperfusion syndrome (PRS) on hyperglycemia occurrence and connecting (C) peptide release, which acts as a surrogate marker for insulin resistance, during the intraoperative period after graft reperfusion in patients undergoing living donor liver transplantation (LDLT) using propensity score (PS)-matching analysis.

### Patients and methods

Medical records from 324 adult patients who underwent elective LDLT were retrospectively reviewed, and their data were analyzed according to PRS occurrence (PRS vs. non-PRS groups) using the PS-matching method. Intraoperative levels of blood glucose and C-peptide were measured through the arterial or venous line at each surgical phase. Hyperglycemia was defined as a peak glucose level >200 mg/dL, and normal plasma concentrations of C-peptide in the fasting state were taken to range between 0.5 and 2.0 ng/mL.

### Results

After PS matching, there were no significant differences in pre- and intra-operative recipient findings and donor-graft findings between groups. Although glucose and C-peptide levels continuously increased through the surgical phases in both groups, glucose and C-peptide levels during the neohepatic phase were significantly higher in the PRS group than in the non-PRS group, and larger changes in levels were observed between the preanhepatic and neohepatic phases. There were higher incidences of C-peptide levels >2.0 ng/mL and peak glucose levels >200 mg/dL in the neohepatic phase in patients with PRS than in those

**Data Availability Statement:** All relevant data are within the manuscript and its Supporting Information files.

**Funding:** The authors received no specific funding for this work.

**Competing interests:** The authors have declared that no competing interests exist.

**Abbreviations:** PRS, postreperfusion syndrome; LDLT, living donor liver transplantation; C-peptide, connecting-peptide; DM, diabetes mellitus; MAP, mean arterial pressure; BMI, body mass index; MELD, model for end-stage liver disease; POD, postoperative day.

without. PRS adjusted for PS with or without exogenous insulin infusion was significantly associated with hyperglycemia occurrence during the neohepatic phase.

## Conclusions

Elucidating the association between PRS and hyperglycemia occurrence will help with establishing a standard protocol for intraoperative glycemic control in patients undergoing LDLT.

## Introduction

In liver transplantation (LT), postreperfusion syndrome (PRS) is a stressful and complex burden that causes severe circulatory and metabolic deterioration; it occurs abruptly during reperfusion of the donated liver graft after unclamping of the portal vein and negatively impacts the early postoperative recovery of patients and grafts [1]. Although surgical techniques, graft preservation care, and anesthetic management have advanced, the incidence of PRS has not decreased significantly, with approximately 30% of patients who undergo LT experiencing PRS; this incidence also does not differ significantly between deceased and living donor LT (LDLT) [2,3]. The underlying pathophysiological mechanism of PRS is not fully understood, but severe hemodynamic instability during PRS has been attributed to the response of the cardiovascular system to the release of vasoactive and inflammatory mediators from the grafted liver, such as tumor necrosis factor-α and interleukins-1, -2 and -8, and to the activation of the immune system of the patient after reperfusion, with the involvement of bradykinin, chemokines, and activated complements [4,5]. PRS, which is a strong hemodynamic and metabolic burden, may play a role in the development of stress hyperglycemia during surgery and anesthesia that eventually leads to adverse clinical outcomes [6].

The connecting (C) peptide, which is co-secreted with insulin from pancreatic $\beta$ cells, is a short polypeptide consisting of 31 amino acids that connects the A- and B-chains of the proinsulin molecule; it may have a metabolic effect and is considered a potential therapeutic target for diabetes mellitus (DM) [7]. Because of its lower degradation rate and negligible hepatic clearance compared to insulin, C-peptide is a cornerstone for the assessment of non-diabetes-associated hypoglycemia and the diagnosis of conditions including insulinoma and factitious hypoglycemia [8]. Higher levels of C-peptide have been related to cardiovascular events and all-cause mortality in non-diabetic patients, because raised C-peptide levels are closely related to the severity of insulin resistance [9]. Additionally, relationships between C-peptide level and parameters of insulin resistance have been observed in critically ill patients, such as those with metabolic disease or type 2 DM [10,11].

PRS is an LT-specific and critical feature that reflects increasing hemodynamic and metabolic loads, but studies to date have not fully investigated the association between PRS and hyperglycemia and insulin resistance. Therefore, we investigated the impact of PRS itself on hyperglycemia occurrence and C-peptide release, by treating PRS as a surrogate marker for insulin resistance, during the intraoperative period after graft reperfusion in patients undergoing LDLT using propensity score (PS)-matching analysis.

## Patients and methods

### Ethical considerations

The Institutional Review Board of Seoul St. Mary's Hospital Ethics Committee approved the protocol for the present study (KC20RISI0176) on April 6, 2020, and the study was performed

in accordance with the principles of the Declaration of Helsinki. The requirement for informed consent was waived because of the retrospective nature of the study.

## Study population

Data for 404 adult patients (aged ≥19 years) who underwent elective LDLT between January 2014 and February 2020 at Seoul St. Mary's Hospital were retrospectively collected from the electronic medical record system. The exclusion criteria included patients who were in a clinically stressed condition before surgery, such as those treated in an intensive care unit because of a need for mechanical ventilation, dialysis, a large inotrope infusion, or blood products transfusion; had chronically uncontrolled (i.e., hemoglobin A1c >6.5%) or type 1 DM due to a poor response to glycemic therapy or deficient capacity of insulin secretion [12,13]; had received an intensive immunosuppressive regimen for an ABO-incompatible LDLT [14]; or had missing laboratory data. Based on the exclusion criteria, 80 patients were not included in the study. A total of 324 adult patients were initially enrolled, and their data were analyzed using the PS-matching method; data from 194 matched patients were included in the final analysis (Fig 1).

## Glycemic control during LDLT

The standardized surgical technique and anesthetic care, including glycemic control, for LDLT have been described in detail previously [15–17]. Briefly, intraoperative glycemic control was performed in accordance with the insulin infusion protocol of Yale University [18]. The targeted range of blood glucose was 80–200 mg/dL during surgery. When the blood glucose level exceeded 200 mg/dL (i.e., hyperglycemia), 2 units of bolus loading and 2 units/h of continuous infusion of regular insulin were applied. If the blood glucose level did not respond to the administered insulin, the infusion rate of regular insulin was doubled. Hypoglycemia was defined as a blood glucose level <80 mg/dL and corrected using 20% or 50% dextrose solution. Intraoperative levels of blood glucose were measured through the arterial line in the preanhepatic, anhepatic, and neohepatic phases; if multiple tests were performed during each phase, the highest glucose level was used in the analysis.

## Measurement of C-peptide level

As part of the intraoperative patient assessment, laboratory variables, including C-peptide level, were measured in all patients undergoing LDLT. Data on C-peptide levels were collected

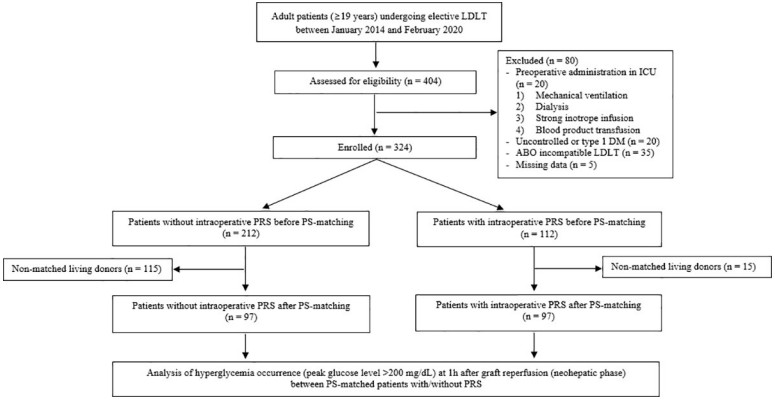

**Fig 1. Flow diagram of the study.** DM: diabetes mellitus, ICU: intensive care unit, LDLT: living donor liver transplantation, PRS: postreperfusion syndrome, PS: propensity score.

during the preanhepatic (i.e., immediately after surgery began) and neohepatic (i.e., at 1 h after graft reperfusion) phases via venous or arterial blood sampling (Clot Activator Tube/BD Vacutainer; Becton, Dickinson and Company, Franklin Lakes, NJ, USA) and measured using an automated chemistry analyzer (Cobas e801; Roche, Basel, Switzerland). Normal plasma concentrations of C-peptide in the fasting state were considered to fall within the range 0.5−2.0 ng/mL [8].

## Definition of PRS

PRS during LDLT was defined as the state in which mean arterial pressure decreased by ≥30% relative to the level at the end of the anhepatic phase, absolute mean arterial pressure was <60 mmHg for at least 1 min within the first 5 min after reperfusion of the grafted liver, or rescue epinephrine (i.e., ≥10 μg) or phenylephrine (i.e., ≥100 μg) infusion was urgently required [1,19].

In our study, the patients were classified into two groups: PRS and non-PRS.

## Primary outcome

Our primary endpoint was hyperglycemia (i.e., peak glucose level >200 mg/dL) occurrence at 1 h after graft reperfusion (i.e., neohepatic phase) in patients with or without PRS. Additionally, blood glucose and C-peptide levels and changes in levels across serial surgical phases were analyzed in both patient groups.

## Perioperative recipient and donor-graft findings

Preoperative recipient findings included age, sex, body mass index (BMI), psoas muscle index, comorbidities, model for end-stage liver disease (MELD) score, hepatocellular carcinoma incidence, hepatic complications, transthoracic echocardiography results based on the 2016 recommendations of the ASE/EACVI [20,21], and laboratory variables. Intraoperative recipient findings included operation time, whether norepinephrine infusion was required, averages of vital signs and laboratory variables, total amount of blood products transfused, hourly fluid infusion, and urine output. Donor-graft findings included age, sex, BMI, graft-to-recipient weight ratio, total ischemic time, fatty change, and hepatic vascular circulation. Postoperatively, we measured glucose levels from the first postoperative day for 1−4 weeks after surgery; early bacteremia status (during the first 4 weeks after surgery) [22]; and new-onset DM developing during the follow-up period [23].

## Statistical analysis

The normality of continuous data was assessed suing the Shapiro–Wilk test. Continuous data are expressed as medians with interquartile ranges (IQRs), and categorial data are expressed as numbers with proportions. We used univariate and multivariate logistic regression to test for associations between demographic factors and the development of postreperfusion syndrome in the entire study population (n = 324). PS-matching analysis was applied to reduce the impact of potential confounding factors on intergroup differences based on PRS. PSs were derived to match patients at a one-to-one ratio using greedy matching algorithms without replacement. Perioperative recipient and donor graft factors were compared using the Mann-Whitney $U$-test and $\chi^2$ test or Fisher's exact test, as appropriate. Wilcoxon's signed-rank sum test and McNemar's test were used to analyze pair-matched data. The association of PRS with hyperglycemia occurrence in the neohepatic phase was evaluated using multivariable logistic regression analysis with adjustment for PS and exogenous insulin infusion. The results are

presented as odds ratios with 95% confidence intervals. All tests were two-sided, and $p < 0.05$ was considered to denote statistical significance. All statistical analyses were performed using R (version 2.10.1; R Foundation for Statistical Computing, Vienna, Austria) and SPSS for Windows software (version 24.0; SPSS Inc., Chicago, IL, USA).

## Results

### Demographic characteristics of patients undergoing LDLT

The study population of 324 patients was predominantly male (72.2%), and the median (IQR) age and BMI were 54 (49–60) years and 24.2 (22.0–26.8) kg/m$^2$, respectively. The most common etiologies of LDLT were as follows: hepatitis B (49.1%), alcoholic hepatitis (25.6%), hepatitis C (7.4%), autoimmune hepatitis (5.9%), hepatitis A (4.3%), drug and toxic hepatitis (1.2%), and cryptogenic hepatitis (6.5%). The median (IQR) MELD score was 15 (7–25) points.

The incidence of PRS was 34.6%, and PRS was associated with the following demographic factors: a higher MELD score, history of an ascites volume $\geq$ 1 L, and normal or grade I diastolic function (S1 Table). However, among patients with diastolic dysfunction (grade II–III), all those with PRS were grade III dysfunction and all those without PRS were grade II dysfunction (S2 Table).

### Comparison of pre- and intra-operative recipient factors and donor-graft factors before and after PS matching

Before PS matching (Table 1), there were significant differences in preoperative recipient findings (i.e., MELD score, hepatocellular carcinoma incidence, ascites ≥1 L, diastolic dysfunction,

**Table 1. Preoperative recipient and donor-graft findings before and after PS-matching analysis.**

| Group | Before PS-matching analysis | | | | After PS-matching analysis | | | |
|---|---|---|---|---|---|---|---|---|
| | non-PRS | PRS | *p* | SD | non-PRS | PRS | *p* | SD |
| n | 212 | 112 | | | 97 | 97 | | |
| *Preoperative recipient findings* | | | | | | | | |
| Age (years) | 54.5 (49.0–60.0) | 54.0 (48.0–61.0) | 0.642 | 0.132 | 55.0 (49.0–61.0) | 54.0 (48.0–61.0) | 0.982 | 0.055 |
| Sex (female) | 55 (25.9%) | 35 (31.3%) | 0.31 | 0.114 | 35 (36.1%) | 31 (32.0%) | 0.544 | -0.089 |
| Body mass index (kg/m$^2$) | 24.6 (22.1–26.8) | 23.6 (21.9–26.6) | 0.347 | -0.005 | 24.3 (21.6–26.7) | 23.8 (22.1–26.9) | 0.998 | 0.080 |
| Psoas muscle index (mm$^2$/m$^2$) | 801.2 (597.0–1137.4) | 787.9 (609.4–943.7) | 0.073 | -0.416 | 758.8 (581.1–939.7) | 786.1 (605.0–899.8) | 0.515 | -0.054 |
| Comorbidity | | | | | | | | |
| Hypertension | 53 (25.0%) | 22 (19.6%) | 0.277 | -0.134 | 16 (16.5%) | 18 (18.6%) | 0.706 | 0.052 |
| Diabetes mellitus | 60 (28.3%) | 33 (29.5%) | 0.826 | 0.025 | 30 (30.9%) | 29 (29.9%) | 0.876 | -0.023 |
| MELD score (points) | 11.6 (5.7–22.4) | 18.6 (11.3–30.0) | <0.001 | 0.501 | 16.5 (7.8–26.0) | 17.6 (10.1–29.4) | 0.29 | 0.149 |
| Hepatocellular carcinoma | 102 (48.1%) | 35 (31.3%) | 0.003 | -0.362 | 35 (36.1%) | 33 (34.0%) | 0.763 | -0.044 |
| Hepatic complications | | | | | | | | |
| Encephalopathy (West-Haven criteria I or II) | 96 (45.3%) | 60 (53.6%) | 0.156 | 0.165 | 50 (51.5%) | 50 (51.5%) | >0.999 | 0.000 |
| Varix | 55 (25.9%) | 31 (27.7%) | 0.737 | 0.039 | 28 (28.9%) | 25 (25.8%) | 0.629 | -0.069 |
| Ascites $\geq$ 1L | 90 (42.5%) | 75 (67.0%) | <0.001 | 0.519 | 63 (64.9%) | 65 (67.0%) | 0.762 | 0.044 |
| Transthoracic echocardiography | | | | | | | | |
| Ejection fraction (%) | 64.4 (62.0–66.0) | 64.4 (62.0–66.1) | 0.489 | 0.083 | 64.0 (62.0–66.4) | 64.4 (62.0–66.2) | 0.518 | 0.084 |
| Diastolic dysfunction ($\geq$ grade II) | 29 (13.7%) | 4 (3.6%) | 0.004 | -0.542 | 7 (7.2%) | 4 (4.1%) | 0.352 | -0.166 |
| Laboratory variables | | | | | | | | |

*(Continued)*

**Table 1.** (*Continued*)

| Group | Before PS-matching analysis | | | | After PS-matching analysis | | | |
|---|---|---|---|---|---|---|---|---|
| | non-PRS | PRS | *p* | SD | non-PRS | PRS | *p* | SD |
| n | 212 | 112 | | | 97 | 97 | | |
| Hematocrit (%) | 30.5 (25.9–37.0) | 27.0 (23.3–32.0) | <0.001 | -0.472 | 27.3 (24.1–33.5) | 27.0 (23.3–32.2) | 0.854 | -0.003 |
| White blood cell count (x $10^9$/L) | 4.6 (3.1–7.6) | 5.6 (3.6–8.9) | 0.071 | 0.092 | 4.9 (3.0–8.3) | 5.5 (3.7–8.5) | 0.191 | 0.156 |
| Neutrophil (%) | 60.5 (51.8–74.2) | 67.3 (56.3–79.5) | 0.003 | 0.185 | 63.9 (56.1–75.4) | 64.7 (55.4–77.0) | 0.6 | 0.020 |
| Lymphocyte (%) | 21.3 (12.6–29.5) | 15.9 (8.9–25.1) | 0.003 | -0.342 | 17.7 (11.7–28.7) | 17.9 (9.2–25.8) | 0.345 | -0.120 |
| Albumin (g/dL) | 3.3 (2.8–3.7) | 2.9 (2.5–3.3) | <0.001 | -0.662 | 3.0 (2.7–3.4) | 2.9 (2.5–3.3) | 0.172 | -0.103 |
| Aspartate aminotransferase (IU/L) | 44.0 (29.0–83.8) | 49.5 (35.3–89.8) | 0.05 | -0.211 | 45.0 (31.0–88.5) | 49.0 (35.0–84.0) | 0.34 | 0.017 |
| Alanine aminotransferase (IU/L) | 29.5 (18.3–59.5) | 29.0 (20.0–54.5) | 0.837 | -0.199 | 27.0 (16.0–53.5) | 29.0 (20.0–52.5) | 0.612 | -0.022 |
| Total bilirubin (mg/dL) | 1.9 (0.8–13.3) | 6.1 (2.0–19.9) | <0.001 | 0.356 | 3.2 (1.1–20.7) | 5.4 (1.3–19.1) | 0.285 | 0.104 |
| Sodium (mEq/L) | 140.0 (136.3–142.0) | 138.0 (134.0–140.8) | <0.001 | -0.264 | 138.0 (135.0–141.0) | 138.0 (134.0–141.0) | 0.447 | -0.060 |
| Calcium (mg/dL) | 8.4 (8.0–8.9) | 8.4 (7.8–9.0) | 0.504 | -0.020 | 8.3 (7.9–8.9) | 8.4 (7.8–9.0) | 0.758 | -0.044 |
| Potassium (mEq/L) | 4.0 (3.6–4.4) | 3.9 (3.5–4.3) | 0.355 | -2.836 | 4.1 (3.6–4.4) | 3.9 (3.5–4.3) | 0.442 | -0.107 |
| Creatinine (mg/dL) | 0.8 (0.7–1.1) | 1.0 (0.7–1.7) | 0.007 | 0.234 | 0.8 (0.6–1.4) | 0.9 (0.7–1.4) | 0.214 | -0.011 |
| Glucose (mg/dL) | 108.5 (93.0–141.8) | 115.0 (93.3–145.0) | 0.565 | 0.093 | 112.0 (92.5–145.5) | 115.0 (92.5–144.0) | 0.816 | 0.049 |
| Platelet count (x $10^9$/L) | 73.5 (50.3–105.0) | 57.5 (42.3–90.5) | 0.004 | -0.339 | 65.0 (45.5–91.0) | 57.0 (40.0–88.0) | 0.37 | -0.009 |
| International normalized ratio | 1.4 (1.2–2.0) | 1.8 (1.3–2.2) | 0.001 | 0.116 | 1.6 (1.3–2.2) | 1.8 (1.3–2.2) | 0.4 | 0.016 |
| Fibrinogen (mg/dL) | 173.4 (117.0–219.0) | 173.4 (118.0–186.5) | 0.34 | -0.156 | 173.0 (110.5–204.5) | 172.0 (113.5–195.0) | 0.71 | 0.049 |
| ***Intraoperative recipient findings*** | | | | | | | | |
| Operation time (min) | 470.0 (418.5–513.8) | 462.5 (410.0–530.0) | 0.82 | 0.095 | 470.0 (407.5–522.5) | 460.0 (410.0–521.0) | 0.722 | 0.013 |
| Requirement of norepinephrine infusion ≥ 0.05 µg/kg/min | 144 (67.9%) | 94 (83.9%) | 0.002 | 0.434 | 73 (75.3%) | 79 (81.4%) | 0.296 | 0.168 |
| Average of vital signs | | | | | | | | |
| Systolic blood pressure (mmHg) | 107.3 (99.1–116.3) | 101.8 (95.3–109.9) | <0.001 | -0.340 | 106.0 (97.8–111.8) | 102.0 (95.4–111.9) | 0.2 | -0.100 |
| Diastolic blood pressure (mmHg) | 56.7 (50.8–62.0) | 54.4 (48.3–58.3) | 0.002 | -0.407 | 54.3 (48.5–60.0) | 54.8 (48.6–58.3) | 0.454 | -0.119 |
| Heart rate (beats/min) | 89.4 (81.1–99.3) | 89.9 (79.6–97.9) | 0.781 | -0.040 | 87.5 (80.5–99.6) | 90.2 (79.0–98.1) | 0.726 | 0.044 |
| Central venous pressure (mmHg) | 9.0 (7.0–11.0) | 8.8 (7.3–11.0) | 0.784 | -0.001 | 9.3 (6.8–11.1) | 9.0 (7.5–10.9) | 0.668 | 0.041 |
| Cardiac index (L/min/m²) | 4.1 (3.5–4.9) | 3.9 (3.2–4.4) | 0.037 | -0.266 | 3.9 (3.3–4.7) | 3.9 (3.5–4.7) | 0.716 | 0.041 |
| Systemic vascular resistance index (dynes-sec/$cm^{-5}$/m²) | 1290.8 (1031.8–1525.9) | 1251.1 (1045.6–1583.4) | 0.876 | -0.080 | 1275.0 (1029.5–1479.6) | 1250.0 (997.0–1565.6) | 0.895 | -0.056 |
| Average of laboratory variables | | | | | | | | |
| Arterial blood pH | 7.36 (7.32–7.39) | 7.34 (7.3–7.38) | 0.033 | 0.095 | 7.34 (7.31–7.39) | 7.34 (7.3–7.37) | 0.45 | -0.004 |
| Hemoglobin (g/dL) | 10.2 (9.1–10.7) | 9.3 (8.4–10.2) | <0.001 | -0.641 | 9.5 (8.7–10.2) | 9.4 (8.4–10.2) | 0.879 | 0.052 |
| Lactate (mmol/L) | 3.8 (3.1–5.2) | 3.6 (3.0–5.1) | 0.287 | 0.009 | 3.8 (3.1–4.8) | 3.5 (3.0–5.1) | 0.446 | 0.017 |
| Brain natriuretic peptide (pg/mL) | 90.4 (41.9–162.7) | 95.5 (53.2–162.7) | 0.261 | 0.073 | 113.6 (64.4–182.3) | 90.2 (53.5–162.7) | 0.201 | 0.027 |
| Total amount of blood product transfusion (unit) | | | | | | | | |
| Packed red blood cell | 6.0 (3.0–10.0) | 12.0 (7.0–19.0) | <0.001 | 0.588 | 10.0 (5.0–12.0) | 10.0 (7.0–16.0) | 0.054 | 0.176 |
| Fresh frozen plasma | 6.0 (4.0–9.0) | 10.0 (6.0–15.0) | <0.001 | 0.530 | 8.0 (4.0–10.0) | 10.0 (5.0–12.0) | 0.057 | 0.156 |
| Single donor platelet | 0.0 (0.0–1.0) | 1.0 (0.0–1.7) | <0.001 | 0.442 | 1.0 (0.0–1.3) | 1.0 (0.0–1.3) | 0.465 | 0.114 |
| Cryoprecipitate | 0.0 (0.0–0.0) | 0.0 (0.0–0.0) | 0.003 | 0.261 | 0.0 (0.0–0.0) | 0.0 (0.0–0.0) | 0.272 | 0.110 |
| Hourly fluid infusion (mL/kg/h) | 11.5 (9.4–15.3) | 15.5 (9.4–21.3) | <0.001 | 0.344 | 13.3 (9.9–17.3) | 14.8 (9.1–18.6) | 0.698 | 0.104 |
| Hourly urine output (mL/kg/h) | 1.4 (0.7–2.2) | 0.8 (0.3–1.8) | <0.001 | -0.231 | 1.1 (0.5–1.8) | 0.8 (0.4–1.6) | 0.12 | -0.088 |
| ***Donor-graft findings*** | | | | | | | | |
| Age (years) | 35.4 (26.3–45.0) | 35.4 (32.0–40.0) | 0.207 | 0.096 | 35.4 (27.5–44.5) | 35.4 (31.0–40.0) | 0.595 | 0.049 |
| Sex (female) | 68 (32.1%) | 23 (20.5%) | 0.028 | -0.284 | 23 (23.7%) | 19 (19.6%) | 0.486 | -0.102 |
| Body mass index (kg/m²) | 20.2 (18.5–21.9) | 20.2 (20.1–22.6) | 0.084 | 0.216 | 20.2 (18.4–22.0) | 20.2 (20.1–22.5) | 0.636 | 0.121 |
| Graft-recipient-weight-ratio (%) | 1.2 (1.0–1.5) | 1.3 (1.1–1.6) | 0.016 | 0.271 | 1.2 (1.0–1.5) | 1.3 (1.1–1.6) | 0.288 | 0.144 |

(*Continued*)

**Table 1.** (Continued)

| Group | Before PS-matching analysis | | | | After PS-matching analysis | | | |
|---|---|---|---|---|---|---|---|---|
| | non-PRS | PRS | *p* | SD | non-PRS | PRS | *p* | SD |
| n | 212 | 112 | | | 97 | 97 | | |
| Total ischemic time (min) | 73.5 (57.0–92.8) | 73.0 (51.0–95.5) | 0.609 | -0.021 | 71.0 (57.0–94.5) | 74.0 (55.5–97.5) | 0.962 | 0.015 |
| Fatty change (%) | 4.9 (1.0–5.0) | 4.9 (1.3–5.0) | 0.05 | 0.231 | 4.9 (0.5–5.0) | 4.9 (1.0–5.0) | 0.127 | 0.140 |
| Hepatic vascular circulation | | | | | | | | |
| Hepatic artery resistive index | 0.64 (0.6–0.7) | 0.64 (0.57–0.68) | 0.155 | -0.148 | 0.64 (0.6–0.71) | 0.64 (0.58–0.69) | 0.256 | -0.159 |
| Portal venous flow (L/min) | 1924.7 (1474.7–2376.7) | 1829.3 (1210.6–2202.2) | 0.102 | -0.163 | 1952.0 (1483.4–2305.6) | 1838.7 (1285.8–2210.0) | 0.307 | -0.078 |

**Abbreviations:** PS, propensity score; PRS, postreperfusion syndrome; MELD, model for end-stage liver disease.

**NOTE:** Values are expressed as median (interquartile) and number (proportion).

hematocrit, neutrophil content, lymphocyte content, albumin content, total bilirubin, sodium content, creatinine content, platelet count, and international normalized ratio), intraoperative recipient findings (i.e., whether ≥0.05 μg/kg/min norepinephrine was required; average systolic and diastolic blood pressure and cardiac index; average arterial blood pH and hemoglobin; total amounts of packed red blood cells, fresh frozen plasma, single donor platelets, and cryoprecipitate transfusions; hourly fluid infusion; and urine output), and donor-graft findings (i.e., sex and graft-to-recipient weight ratio) between groups; however, after PS matching, there were no significant differences in pre- and intra-operative recipient findings or donor-graft findings between groups.

## Comparison of glucose and C-peptide levels in PS-matched patients with or without PRS

As listed in Table 2, although glucose and C-peptide levels continuously increased through the surgical phases in both groups, levels in the neohepatic phase were significantly higher in the PRS group than in the non-PRS group, and changes in levels from the preanhepatic to the neohepatic phase were larger. PRS patients required insulin infusion more frequently than did non-PRS patients.

As shown in Fig 2, there was a higher incidence of C-peptide level >2.0 ng/mL in the neohepatic phase in the PRS group than in the non-PRS group (70.1% vs. 54.6%, *p* = 0.026).

Additionally, as shown in Fig 3, there was a higher incidence of peak glucose level >200 mg/dL in the neohepatic phase in the PRS group than in the non-PRS group (84.5% vs. 59.8%, *p* < 0.001). Between postoperative days 2 and 7, the incidence of peak glucose level >200 mg/dL was also higher in the PRS group than in the non-PRS group.

## Association of PRS with hyperglycemia occurrence in the neohepatic phase in PS-matched patients

As listed in Table 3, PRS adjusted for PS with or without exogenous insulin infusion was significantly associated with hyperglycemia occurrence in the neohepatic phase.

## Comparison of other outcomes in PS-matched patients with and without PRS

Although statistical significance was not attained, new-onset diabetes mellitus and early bacteremia were more frequent in the PRS than the non-PRS group (S3 Table). S4 Table shows

**Table 2. Comparison of intraoperative glucose and C-peptide levels and requirement of insulin infusion in PS-matched patients with/without PRS.**

| Group | non-PRS | PRS | p |
|---|---|---|---|
| n | 97 | 97 | |
| Glucose level (mg/dL) | | | |
| at the preanhepatic phase | 116.0 (102.0–146.0) | 122.0 (106.0–142.0) | 0.698 |
| at the anhepatic phase | 135.0 (113.0–164.5) | 144.0 (116.5–183.0) | 0.305 |
| at the neohepatic phase | 208.0 (171.5–233.0)[†††] | 242.0 (211.5–267.5)[†††] | <0.001 |
| Change of glucose level (%) | | | |
| from the preanhepatic to neohepatic phases | 162.4 (133.8–212.9) | 200.0 (160.8–246.3) | <0.001 |
| C-peptide level (ng/mL) | | | |
| at the preanhepatic phase | 2.19 (1.38–3.24) | 2.34 (1.55–3.24) | 0.575 |
| at the neohepatic phase | 2.25 (1.54–3.95)[††] | 3.23 (1.79–5.06)[†††] | 0.004 |
| Change of C-peptide level (%) | | | |
| from the preanhepatic to neohepatic phases | 129.71 (71.29–177.04) | 167.52 (90.85–247.62) | 0.04 |
| Total insulin infusion (unit) | 10.0 (3.0–22.3) | 15.0 (10.0–21.5) | 0.02 |

**Abbreviations:** PS, propensity score; PRS, postreperfusion syndrome.

[†]$p<0.05$

[††]$p\leq0.01$ and

[†††]$p\leq0.001$ based on the level at the preanhepatic phase.

**NOTE:** Values are expressed as median and interquartile.

the incidence rates of pre-transplant DM and post-transplant new-onset DM by LDLT etiology in PS-matched patients. In patients with overt pre-transplant DM (n = 59), the most common etiology was alcoholic hepatitis (42.4%), while in those with post-transplant new-onset DM (n = 38) it was hepatitis B infection (44.7%).

## Discussion

The main findings of our study are that an increase of stress factors related to PRS may worsen insulin resistance, as measured intraoperatively by the C-peptide level, and subsequently result in peak glucose levels of >200 mg/dL during the neohepatic phase and the first week after

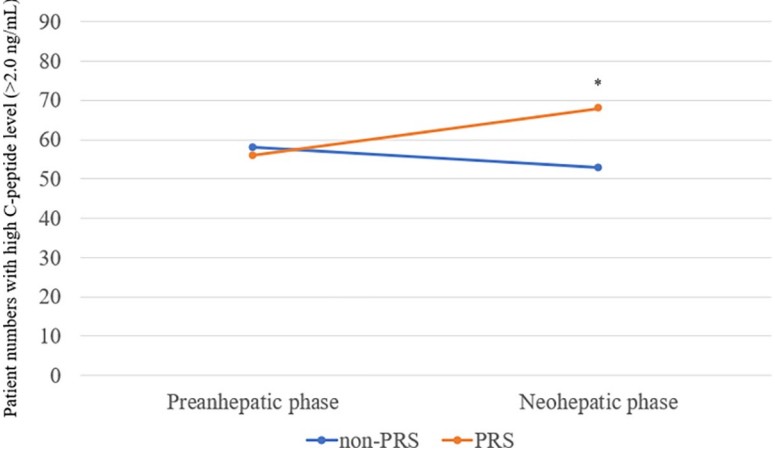

**Fig 2. Comparison of occurrence of a high level of connecting peptide (>2.0 ng/mL) between propensity score-matched patients with or without postreperfusion syndrome (PRS).** $^*p < 0.05$.

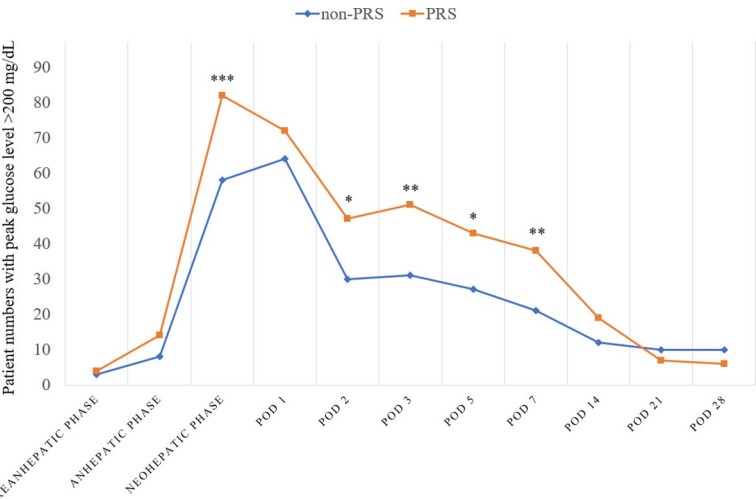

**Fig 3. Comparison of intra- and postoperative hyperglycemia occurrence (peak glucose level > 200 mg/dL) between propensity score-matched patients with or without postreperfusion syndrome (PRS).** $^*p < 0.05$, $^{**}p \leq 0.01$, and $^{***}p \leq 0.001$. POD, postoperative day.

surgery. In PS-matched patients, hyperglycemia occurrence in the neohepatic phase in PRS patients was approximately three-fold higher than that in patients without PRS. The stress insult with PRS weakened control of the glucose level, leading to a higher likelihood of needing exogenous insulin infusion during surgery.

Stress hyperglycemia occurs in 38% of patients undergoing surgery, and those patients who newly developed hyperglycemia have been found to have a significantly higher rate of in-hospital mortality (16%) compared to patients with normoglycemia (1.7%) [6]. The combined activation of hormonal and inflammatory responses, which may be part of the surgical stress response, leads to insulin resistance, the result of a decrease in insulin sensitivity that is characterized by an increase in the production of endogenous hepatic glucose, a decrease in the uptake of peripheral glucose, and an increase in the breakdown of protein. Additionally, surgical tissue injury, pain, the withholding of food and fluids, and poor mobilization cause further losses of insulin sensitivity and an increased catabolic state [24]. Approximately 30−60% of cirrhotic patients exhibit metabolic impairment of blood glucose (hepatogenic diabetes) that reflects insulin resistance in the muscle, fat, and liver, as well as increasing secretion of insulin (hyperinsulinemia) [25]. Hyperglycemia with insulin resistance is strongly associated with endothelial dysfunction, which can aggravate hepatic vascular resistance and portal hypertension [26], and with platelet aggregation and thrombus formation [27]. The major and adverse features of LT, such as extensive tissue dissection, liver and large vessel manipulations, blood product transfusion, and catecholamine infusion, progressively exacerbate diabetogenic features, leading to severe hyperglycemia occurrence after reperfusion of liver grafts and postoperatively, which increases the risk of postoperative complications, such as surgical site

**Table 3. Association of PRS with occurrence of hyperglycemia (>200 mg/dL) at the neohepatic phase in PS-matched patients.**

| | ß | Odds ratio | 95% confidence interval | p |
|---|---|---|---|---|
| PRS adjusted PS | 1.302 | 3.676 | 1.855−7.284 | <0.001 |
| PRS adjusted PS and exogenous insulin infusion | 1.254 | 3.504 | 1.751−7.014 | <0.001 |

**Abbreviations:** PRS, postreperfusion syndrome; PS, propensity score.

infection, delayed wound healing, impaired immune function, and increased length of stay [28,29].

Our study results suggest that PRS may be an independent factor with negative impacts on intraoperative glycemic control that causes pronounced systemic insulin insensitivity and pancreatic hypersecretion of insulin. Although the underlying mechanism of PRS related to hyperglycemia is uncertain, ischemia–reperfusion injuries in grafts and patients involve biochemical and cellular changes that produce pro-inflammatory cytokines and oxygen free radicals as well as activate the complement system, which leads to an inflammatory response that is mediated by neutrophil and platelet interactions associated with swelling of the endothelium, vasoconstriction, leukocyte sedimentation, and hemoconcentration [30–32]. The production of inflammatory mediators may contribute to PRS and cause a profound local inflammatory response, which eventually leads to systemic inflammatory response syndrome, activation of hepatic gluconeogenesis, and peripheral insulin resistance [33]. Although PRS seems to occur in an unpredictable manner, understanding the risk factors that are significantly associated with it, such as hyperkalemia, hypothermia, old age of donor, large blood product transfusion, prolonged ischemic time, and ventricular diastolic dysfunction, is essential because effective treatment strategies can be identified for patients at risk of imminent hemodynamic and metabolic collapse [2,34,35].

There were some limitations in our study. First, we were not able to directly measure pancreatic $\beta$ cell function or severity of insulin resistance before surgery. However, because we excluded patients with type 1 or uncontrolled DM from the analysis, patients in our study might have had an acceptable ability to secrete insulin from the pancreas in response to metabolic stimuli. Previous studies have suggested a significant correlation between C-peptide level and degree of insulin resistance [9–11]. Second, we did not investigate the association between PRS related to hyperglycemia and new occurrence of DM as a long-term postoperative complication. Although resolution of hyperglycemia is expected after successful LT upon good recovery of the liver graft's function [36], further study is required to investigate the effects of intraoperative PRS-induced hyperglycemia on the postoperative occurrence of overt DM.

## Conclusions

Intraoperative stress hyperglycemia is a common clinical issue due to a transient decrease in insulin responsiveness; it may persist for days or weeks after major surgery. Various factors influence the timing, severity, and duration of stress hyperglycemia, and patients without established DM who develop stress hyperglycemia are at higher risk of poor outcomes. Cirrhotic patients exhibit features of hepatic diabetes, which manifests as peripheral insulin resistance, hyperinsulinemia, and particularly, PRS, which acts as an LT-specific stress factor that may lead to overt hyperglycemia, with the peak glucose level occurring after graft reperfusion. Because there are no specific guidelines, elucidating the association between PRS and hyperglycemia occurrence would help with establishing a standard protocol for intraoperative glycemic control in patients undergoing LDLT.

## Supporting information

**S1 Table. Associations of demographic factors with postreperfusion syndrome in the entire study population (n = 324).**
(DOCX)

**S2 Table. Comparison of diastolic dysfunction before surgery between all patients with/ without PRS (n = 324).**
(DOCX)

**S3 Table. The rates of new-onset diabetes mellitus during the follow-up period and early bacteremia during the first 4 weeks postoperatively in PS-matched patients with and without PRS.**
(DOCX)

**S4 Table. The rates of pre-transplant DM and post-transplant new-onset DM by LDLT etiology in PS-matched patients.**
(DOCX)

## Acknowledgments

All authors thank Eunju Choi, Hyeji An and Hyunsook Yoo (Anesthesia Nursing Unit, Seoul St. Mary's Hospital, College of Medicine, The Catholic University of Korea, Seoul, Republic of Korea) for participation of our study.

## Author Contributions

**Conceptualization:** Min Suk Chae.

**Data curation:** Junghee Choi, Sujin Lim, Ho Joong Choi, Jaesik Park, Sang Hyun Hong, Chul Soo Park, Jong Ho Choi, Min Suk Chae.

**Formal analysis:** Sumin Chae, Junghee Choi, Sujin Lim, Ho Joong Choi, Jaesik Park, Sang Hyun Hong, Chul Soo Park, Jong Ho Choi, Min Suk Chae.

**Investigation:** Min Suk Chae.

**Methodology:** Sumin Chae, Min Suk Chae.

**Supervision:** Min Suk Chae.

**Validation:** Min Suk Chae.

**Visualization:** Min Suk Chae.

**Writing – original draft:** Sumin Chae, Min Suk Chae.

**Writing – review & editing:** Min Suk Chae.

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
