## [Decision Letter · Decision Letter 0]

28 Oct 2020

PONE-D-20-29605

Stress burden related to postreperfusion syndrome may aggravate hyperglycemia with insulin resistance during living donor liver transplantation: a propensity score-matching analysis

PLOS ONE

Dear Dr. Chae,

Thank you for submitting your manuscript to PLOS ONE. After careful consideration, we feel that it has merit but does not fully meet PLOS ONE’s publication criteria as it currently stands. Therefore, we invite you to submit a revised version of the manuscript that addresses the points raised during the review process.

Please revise accordingly.

We look forward to receiving your revised manuscript.

Kind regards,

Academic Editor

PLOS ONE

Journal Requirements:

Reviewers' comments:

Reviewer's Responses to Questions

**Comments to the Author**

1. Is the manuscript technically sound, and do the data support the conclusions?

Reviewer #1: Yes

Reviewer #2: Yes

2. Has the statistical analysis been performed appropriately and rigorously? 

Reviewer #1: Yes

Reviewer #2: Yes

3. Have the authors made all data underlying the findings in their manuscript fully available?

Reviewer #1: No

Reviewer #2: Yes

4. Is the manuscript presented in an intelligible fashion and written in standard English?

Reviewer #1: Yes

Reviewer #2: Yes

5. Review Comments to the Author

Reviewer #1: The authors revealed the findings that post-reperfusion syndrome could have an pivotal role in glucose homeostasis during operation but improved 7 days after transplant.

Major concerns.

1. please provide the data of relationship between post-reperfusion syndrome and long-term denovo DM and infection to show clinical significant impact.

2.From pre-operative demographics, please analyse the risk factors, which will predict the occurrence of post-reperfusion syndrome

Reviewer #2: Dear authors, thanks for the opportunity to review your article.

In general, the study could be comprehensive and interesting for readers.

There is only one comment to the authors:

<< As known, hepatitis C virus cold be related to type 2 DM. I suggest that the authors should analyze underlying liver disorders, such as alcoholic or hepatitis C /B liver diseases.

6. PLOS authors have the option to publish the peer review history of their article (what does this mean?). If published, this will include your full peer review and any attached files.

Reviewer #1: No

Reviewer #2: No

---

## [Author Response · Author response to Decision Letter 0]

18 Nov 2020

Point-by-point response letter

Academic Editor

PLoS ONE

Dear Dr. Chen,

We thank you for the opportunity to revise our manuscript (PONE-D-20-29605) for publication in PLoS ONE. We also thank the reviewers for their insightful comments, which have helped us to considerably improve our manuscript. Our point-by-point responses to the reviewers’ comments are presented below, along with our revisions and the revised manuscript.

The authors have no conflicts of interest to declare, and no funding was provided for this work. The authors confirm that neither the manuscript nor any part of its contents is currently under consideration by, nor has been published in, any other journal. The English in this document has been checked by at least two professional editors, both native speakers of English. For a certificate, please see: http://www.textcheck.com/certificate/I2Wujv

Reviewer #1: The authors revealed the findings that post-reperfusion syndrome could have an pivotal role in glucose homeostasis during operation but improved 7 days after transplant.

Major concerns.

1. please provide the data of relationship between post-reperfusion syndrome and long-term denovo DM and infection to show clinical significant impact.

Response: As the reviewer suggested, we compared the rate of development of new-onset diabetes mellitus during the follow-up period, and that of early bacteremia during the first 4 postoperative weeks, between PS-matched patients with and without PRS. Although statistical significance was not attained, new-onset diabetes mellitus and early bacteremia were more frequent in the PRS than the non-PRS group (Table S3). We have added information on new-onset DM and early bacteremia to the “Patients and Methods” section (subheading: “Perioperative recipient and donor-graft findings”; pages 10 and 11) and the “Results” section (subheading: “Comparison of other outcomes in PS-matched patients with and without PRS”; page 23). 

2. From pre-operative demographics, please analyse the risk factors, which will predict the occurrence of post-reperfusion syndrome

Response: As the reviewer suggested, we analyzed the associations between demographic factors and the postreperfusion syndrome rate in the entire study population (n = 324) using univariate and multivariate logistic regression, as described in the “Statistical analysis” section (page 11) and the “Results” section (subheading: “Demographic characteristics of patients undergoing LDLT”; pages 12 and 13). In the univariate logistic regression, the psoas muscle index, MELD score, hepatocellular carcinoma status, ascites volume ≥ 1 L, diastolic dysfunction, hematocrit, platelet and lymphocyte counts, and albumin, total bilirubin, neutrophil and creatinine levels were associated with the development of postreperfusion syndrome (p≤0.1). In multivariate logistic regression, a higher MELD score, history of an ascites volume ≥ 1 L, and normal or grade I diastolic function were independently associated with the development of postreperfusion syndrome (Table S1; page 12). However, the association with diastolic dysfunction should be interpreted cautiously because clinical diastology (performed as suggested by the 2016 ASE/EACVI recommendations in patients who had undergone LDLT) reduced the prevalence of overt diastolic dysfunction and increased that of normal diastolic function (see “Perioperative recipient and donor-graft findings” on page 9). Among the patients with diastolic dysfunction (grade II–III), all those with PRS were grade III dysfunction and all those without PRS were grade II dysfunction (Table S2; pages 12 and 13).

Reviewer #2: Dear authors, thanks for the opportunity to review your article.

In general, the study could be comprehensive and interesting for readers.

There is only one comment to the authors:

<< As known, hepatitis C virus cold be related to type 2 DM. I suggest that the authors should analyze underlying liver disorders, such as alcoholic or hepatitis C /B liver diseases.

Response: As the reviewer suggested, we tested for an association between disease etiology and pre-transplant, new-onset diabetes mellitus in PS-matched patients (Table S4; page 23); the results are provided in the “Results” section (“Comparison of other outcomes in PS-matched patients with or without PRS” subsection; page 23). In patients with pre-transplant DM (n = 59), the most common etiology was alcoholic hepatitis (42.4%), while in post-transplant new-onset DM patients (n = 38) it was hepatitis B infection (44.7%). These findings may be caused by that hepatitis B is more common cause of end-stage liver disease, that subsequently lead to LDLT in Asian community, rather than hepatitis C (DOI: 10.1002/hep.27969). 

We believe that our manuscript has been greatly improved as a result of the revision process. We hope that the revised manuscript is now suitable for publication in PLoS ONE.

Yours sincerely,

Min Suk Chae, MD, PhD

---

## [Decision Letter · Decision Letter 1]

30 Nov 2020

Stress burden related to postreperfusion syndrome may aggravate hyperglycemia with insulin resistance during living donor liver transplantation: a propensity score-matching analysis

PONE-D-20-29605R1

Dear Dr. Chae,

We’re pleased to inform you that your manuscript has been judged scientifically suitable for publication and will be formally accepted for publication once it meets all outstanding technical requirements.

Kind regards,

Academic Editor

PLOS ONE

Additional Editor Comments (optional):

Reviewers' comments:

Reviewer's Responses to Questions

**Comments to the Author**

1. If the authors have adequately addressed your comments raised in a previous round of review and you feel that this manuscript is now acceptable for publication, you may indicate that here to bypass the “Comments to the Author” section, enter your conflict of interest statement in the “Confidential to Editor” section, and submit your "Accept" recommendation.

Reviewer #1: All comments have been addressed

Reviewer #2: All comments have been addressed

Reviewer #3: All comments have been addressed

2. Is the manuscript technically sound, and do the data support the conclusions?

Reviewer #1: Yes

Reviewer #2: Yes

Reviewer #3: Yes

3. Has the statistical analysis been performed appropriately and rigorously? 

Reviewer #1: Yes

Reviewer #2: Yes

Reviewer #3: Yes

4. Have the authors made all data underlying the findings in their manuscript fully available?

Reviewer #1: Yes

Reviewer #2: Yes

Reviewer #3: No

5. Is the manuscript presented in an intelligible fashion and written in standard English?

Reviewer #1: Yes

Reviewer #2: Yes

Reviewer #3: Yes

6. Review Comments to the Author

Reviewer #1: All the questions were addressed and the study provided the evidence for further investigation for post-transplant hypoperfusion syndrome.

Reviewer #2: The authors had completely answered my question. I do not have any more question in the concerns about research ethics, or publication ethics .

Reviewer #3: (No Response)

7. PLOS authors have the option to publish the peer review history of their article (what does this mean?). If published, this will include your full peer review and any attached files.

Reviewer #1: No

Reviewer #2: No

Reviewer #3: No

---

## [Editor Report · Acceptance letter]

2 Dec 2020

PONE-D-20-29605R1 

Stress burden related to postreperfusion syndrome may aggravate hyperglycemia with insulin resistance during living donor liver transplantation: a propensity score-matching analysis 

Dear Dr. Chae:

I'm pleased to inform you that your manuscript has been deemed suitable for publication in PLOS ONE. Congratulations! Your manuscript is now with our production department. 

Kind regards, 

on behalf of

Dr. Robert Jeenchen Chen 

Academic Editor

PLOS ONE